# Influence of Body Mass Index and Duration of Disease on Chromosome Damage in Lymphocytes of Patients with Diabetes

**DOI:** 10.3390/life13091926

**Published:** 2023-09-16

**Authors:** Laura Šiaulienė, Jūratė Kazlauskaitė, Dalia Jurkėnaitė, Žydrūnė Visockienė, Juozas R. Lazutka

**Affiliations:** 1Vilnius University Life Sciences Center, Saulėtekio Al. 7, LT-10257 Vilnius, Lithuania; jurate.mierauskiene@gf.vu.lt (J.K.); daliajurknait@gmail.com (D.J.); 2Vilnius University Hospital Santaros Klinikos, Santariškių St. 2, LT-08661 Vilnius, Lithuania; zydrune.visockiene@santa.lt; 3Faculty of Medicine, Vilnius University, M. K. Čiurlionio St. 21, LT-03101, Vilnius, Lithuania

**Keywords:** body mass index, diabetes, diabetes duration, chromosome damage

## Abstract

It is well-established that patients with diabetes mellitus (DM) have a higher incidence of several types of cancer. The precise mechanisms of this association are still unknown, but obesity and chronic inflammation-induced reactive oxygen species (ROS) are thought to be the main risk factors. ROS may produce different DNA damage, which could eventually lead to cancer. The main objective of this study was to evaluate the relation of chromosome aberrations (CA) with disease status, demographics, and clinical parameters in 33 subjects with type 1 DM (T1DM), 22 subjects with type 2 DM (T2DM), and 21 controls. CAs were analyzed in cultured peripheral blood lymphocytes and subdivided into chromatid (CTA)- and chromosome (CSA)-type aberrations. Compared with controls, higher levels of CTAs and CSAs were observed in T1DM (*p* = 0.0053 and *p* = 0.0203, respectively) and T2DM (*p* = 0.0133 and *p* = 0.00002, respectively). While there was no difference in CTAs between T1DM and T2DM, CSAs were higher in T2DM (*p* = 0.0173). A significant positive association between CTAs and disease duration (r_s_ = 0.2938, *p* = 0.0099) and between CSAs and disease duration (r_s_ = 0.4306, *p* = 0.0001), age (r_s_ = 0.3932, *p* = 0.0004), and body mass index (BMI) (r_s_ = 0.3502, *p* = 0.0019) was revealed. After multiple regression analysis, duration of disease remained significant for CTA, CSA, and CAs (*p* = 0.0042, *p* = 0.00003, and *p* = 0.00002, respectively). For CSA, BMI and the use of statins were the other important confounding variables (*p* = 0.0105 and *p* = 0.0763). Thus, this study demonstrated that both T1DM and T2DM patients had a higher number of all types of aberrations than controls, which increases with the prolonged disease duration. Higher BMI was associated with a higher frequency of CSA. The use of statins might be beneficial for reducing chromosome damage, but further investigations are needed to confirm this association.

## 1. Introduction

Diabetes mellitus (DM) is a metabolic disorder characterized by impaired insulin secretion and/or insulin action resulting in hyperglycemia [1]. The most common are type 1 (T1DM) and type 2 (T2DM) diabetes, which constitute about 5–10% and 90–95% of total diabetes cases, respectively [2]. Both types manifest with sustained hyperglycemia, which is associated with an increased risk of cardiovascular disease and microvascular complications, such as diabetic retinopathy, nephropathy, and neuropathy. According to the International Diabetes Federation (IDF), DM is one of the leading causes of premature death and was responsible for 6.7 million deaths in 2021 [3].

The progression of diabetes complications is associated with several pathogenetic pathways. One of the most recognized is the overproduction of reactive oxygen species (ROS) and the consequent increment of oxidative stress, which causes oxidative damage to different cellular molecules such as carbohydrates, lipids, and DNA [4]. Hyperglycemia in both types of diabetes induces the production of ROS via multiple pathogenetic pathways, whereas other comorbidities, more commonly associated with T2DM and chronic inflammation, such as obesity, dyslipidemia, and hypertension, might further enhance the progression of oxidative damage [5].

Epidemiological studies report that patients with diabetes have a higher incidence of several types of cancer, such as hepatic, pancreatic, endometrial, colorectal, bladder, and breast cancers [6,7]. Excess cancer risk and higher cancer-related mortality are common in both T1DM and T2DM [8]. The precise mechanisms of this association are still unknown, but hyperinsulinemia, hyperglycemia, obesity, and chronic inflammation-induced ROS are thought to be the main risk factors [9]. Inflammation may produce different DNA damage, including single- and double-strand breaks [10] that could lead to chromosome aberrations [11] or genomic instability [12]. On the other hand, chromosome aberrations are well-known predictors of cancer risk [13], with chromosome-type aberrations being more strongly predictive than chromatid-type aberrations [14].

The first studies on chromosome damage in patients with diabetes were published more than four decades ago [15,16]. Although, since that time, several dozens of papers on DNA damage in diabetes were published [17,18,19], most of them used Comet assay or micronuclei (MN) as biomarkers, and only a few used chromosome aberrations.

Thus, our study aims to analyze chromosome aberrations (CA) in peripheral blood lymphocytes from patients with T1DM and T2DM and control subjects. Specific types of chromosome aberrations (chromatid breaks, chromatid exchanges, chromosome breaks, and chromosome exchanges) were recorded separately, enabling us to evaluate their relation with disease status and different demographic and clinical parameters using univariate and multivariate models.

## 2. Materials and Methods

### 2.1. Study Subjects

A cross-sectional study was conducted at Vilnius University Hospital Santaros Klinikos (VUHSK) Endocrinology Center from August 2019 to June 2022. Subjects with T1DM, T2DM, and controls without DM, aged 18 years and older, were included in the study except for those with malignant, severe uncontrolled systemic diseases, mental illnesses, pregnancy, or breastfeeding. A total of 33 T1DM, 22 T2DM, and 21 controls were enrolled. Diagnosis of diabetes was established according to medical records or, for subjects with newly diagnosed diabetes, according to World Health Organization diagnostic criteria [20]. The control group consisted of VUHSK staff/workers, their relatives, and Vilnius University students. To exclude the diagnosis of diabetes, the control group underwent morning fasting venous plasma glucose (FPG) testing, and those with FPG less than 6 mmol/L were enrolled in the study.

Medical history of all subjects was collected from the interview by a trained endocrinologist and from medical records. Heart rate, systolic and diastolic blood pressure (SBP and DBP), body weight (BW), height, and body mass index (BMI) were measured during physical examination. Hypertension was diagnosed if blood pressure was at or above 140/90 mmHg for several measurements or if the patient was using antihypertensive treatment.

Subjects with diabetes were tested for blood lipids, and dyslipidemia was diagnosed according to the European Society of Cardiology Guidelines [21] or in case of statin use. Glycated hemoglobin (HbA1c) data were collected from medical records, and poor glycemic control was considered if at least two recent HbA1c values were more than 7 mmol/mol.

Subjects with diabetes underwent evaluation for diabetic polyneuropathy (DPN) using Neuropathy Symptom Score (NSS) and Neuropathy Disability Score (NDS), as described previously [22]. Cardiac autonomic neuropathy (CAN) was assessed using cardiovascular autonomic reflex tests (CARTs) [23]. The diagnosis of diabetic nephropathy was based on information from medical records.

The study was conducted according to the declaration of Helsinki and was approved by Vilnius Regional Biomedical Research Ethics Committee (registry number 2019/6-1146-635). All participants gave written consent.

### 2.2. Cytogenetic Procedures

Peripheral blood lymphocytes were obtained via venipuncture, and heparinized whole blood was diluted at a ratio of 1:15 with RPM1 1640 media supplemented with 12% heat-inactivated newborn calf serum, 7.8 μg/mL of phytohemagglutinin, and 50 μg/mL of gentamicin. All reagents were purchased from Sigma (St. Louis, MO, USA). Cells were cultured in sterile bottles for 48 h at 37 °C in 5% CO_2_ atmosphere. Colchicine was added into the culture for the last 3 h at a final concentration of 0.6 μg/mL. Cultures were harvested with centrifugation, followed by 30 min hypotonic treatment (0.075M KCl at 37 °C) and three periods of fixation in ethanol–glacial acetic acid (3:1)). Flame-dried slides were prepared and stained for 10 min with Giemsa stain diluted with distilled water (1:9).

Chromosome aberrations were scored on coded slides using bright-field microscope Nikon Eclipse E200 (Nikon, Tokyo, Japan). At least 200 cells per individual were scored. The cells were selected for centromere number no less than 44, good morphology, and clear staining. The best of duplicate cultures was used first, and the other used if necessary to obtain a sufficient number of cells. Each aberration was confirmed by at least two scorers.

Aberrations were scored as individual types, but for statistical analysis, were grouped as listed below.

Chromatid-type aberrations (CTA): chromatid breaks (CTB)—discontinuities more than the width of a chromatid or discontinuities equal to the width of a chromatid with displacement from the chromatid axis and without visible connecting material; chromatid exchanges (CTE)—triradials, symmetrical and asymmetrical quadriradials, chromatid inter- and intrachanges, as well as isochromatid breaks with sister union.

Chromosome-type aberrations (CSA): chromosome breaks (CSB)—terminal breaks without sister union, acentric fragments in the absence of an aberration that could have generated the fragment, and interstitial deletions (acentric rings and double minutes); chromosome exchanges (CSE)—polycentric chromosomes (dicentrics, tricentrics, etc.), ring chromosomes, translocations, and inversions. When an acentric fragment was found accompanying a polycentric or ring chromosome, the combination was scored as one aberration.

Representative images of each group of aberrations are shown in Figure 1.

### 2.3. Statistical Analysis

Categorical variables were characterized using frequencies, and a chi-square test was used to compare the frequencies of nominal variables. Quantitative variables were expressed as mean, and a comparison between two groups was made using the Mann–Whitney U-test and between three groups using the Kruskal–Wallis test. For multiple regression analysis, chromosome aberration data were transformed through average square root transformation: *Y* = 0.5[(*X*)^0.5^ + (*X* + 1)^0.5^], where *X* is the number of chromosome aberrations per 100 cells and *Y* is the transformed variable. This transformation was shown to be effective in stabilizing dispersion [24]. All independent variables listed in Table 1 were included into the primary multiple regression models. The best subset of independent variables was selected with stepwise multiple regression based on p and variance inflation factor (VIF) values. Final models were selected based on R-square, coefficient of multiple correlation, and goodness of fit criteria. All models were validated using residual normality, lack of multicollinearity, and the priori power of >0.8. All statistical calculations were performed using the online statistical calculator Statistics Kingdom [25].

## 3. Results

We studied chromosome aberrations in peripheral blood lymphocytes obtained from 33 patients with T1DM, 22 with T2DM, and 21 subjects without diabetes. The demographic and clinical data of study subjects are presented in Table 1. The proportion of males was significantly higher in T1DM as compared with the control group. Patients with T2DM were statistically significantly older, more often had relatives with diabetes, and more often were diagnosed with obesity, dyslipidemia, and arterial hypertension (AH), leading to a higher prevalence of antihypertensive drugs and statin use compared with T1DM and controls. However, there was no difference in diabetes duration and microvascular complications between subjects with diabetes. Seventy-nine percent of T1DM and sixty-eight percent of T2DM patients were treated with insulin, and eighty-two percent of T2DM patients were on metformin.

Multivariate methods of statistical analysis were needed due to the somewhat imbalanced demographic and clinical parameters of the studied groups of subjects. However, we first performed a series of univariate tests to better select dependent and independent variables for multivariate analysis.

As a first step, we analyzed the frequency of CTB, CTE, CSB, CSE, and total frequency of aberrations per 100 cells in different groups of studied subjects using the Kruskal–Wallis test (Table 2). Clear inter-group variability in mean values was found for CTB (*p* = 0.0059), CSB (*p* = 0.0002), and total CAs (*p* = 0.0002). Differences in mean values of CTE were not significant (*p* = 0.8981), while in the case of CSE, borderline significance was found (*p* = 0.0503). A statistically significant (*p* = 0.0159, Mann–Whitney U-test) difference in CSE values was found when we compared control subjects with pooled group of patients with diabetes. However, the frequencies of both CTE and CSE were too low for any further meaningful statistical analysis—these types of aberrations were not found at all in 15 control subjects, 15 T1DM, and 11 T2DM patients. Thus, for subsequent analysis, CTB and CTE were combined into the CTAs (chromatid-type aberrations) group, while the CSB and CSE were combined into the CSAs (chromosome-type aberrations) group.

Figure 2 shows the frequencies of CTAs and CSAs in different groups of studied subjects. Higher levels of CTAs were observed in both T1DM and T2DM groups as compared with controls (*p* = 0.0053 and *p* = 0.0133, respectively; Mann–Whitney U-test), while there was no difference in CTA mean values between the two groups of patients with diabetes (*p* = 0.9347). For the CSA, mean values in T1DM and T2DM patients were significantly higher than those in controls (*p* = 0.0203 and *p* = 0.00002, respectively), but in T2DM patients they were also higher than in T1DM patients (*p* = 0.0173).

We further analyzed how different demographic and clinical parameters could influence the frequency of chromosome aberrations. Table 3 shows the distribution of CTAs, CSA, and total aberrations in the studied groups of individuals according to their categorical demographic and clinical variables. However, no statistically significant associations were found, and the standardized effect sizes were small. On the contrary, such variables as BMI, disease duration, and age had quite clear influence on some categories of chromosome aberrations. For CTAs, there was a statistically significant positive correlation with duration of disease (r_s_ = 0.2938, *p* = 0.0099; Figure 3), but no significant correlation was found with age (r_s_ = 0.1754, *p* = 0.1295) and BMI (r_s_ = 0.0391, *p* = 0.7370). For CSA, there was a statistically significant positive correlation with all three variables—duration of disease (r_s_ = 0.4306, *p* = 0.0001; Figure 3), age (r_s_ = 0.3932, *p* = 0.0004), and BMI (r_s_ = 0.3502, *p* = 0.0019; Figure 4). However, it is evident that subject‘s age and disease duration are highly correlated variables (r_s_ = 0.4306, *p* = 0.0001), so we should consider this during further calculations.

Multiple regression was introduced as the final step of the analysis. Since the mean values of chromosome aberrations are not normally distributed (Figure 2 and Figure 3), the frequency of aberrations (CTA, CSA, and total CA) per 100 cells was transformed using the formula *Y* = 0.5[(*X*)^0.5^ + (*X* + 1)^0.5^], where *X* is the number of chromosome aberrations per 100 cells, and *Y* is the transformed variable. The results of the D’Agostino–Pearson test indicated that there is no significant difference from the normal distribution for all three transformed variables—CTA_t_, CSA_t_, and CA_t_ (*p* = 0.6010, *p* = 0.3571, and *p* = 0.4813, respectively). We used stepwise multiple regression to select the best subset of independent variables. The best model for the transformed frequency of CSA, showing the highest R-square, goodness of fit, and coefficient of multiple correlation values, included variables such as BMI, duration of disease, and use of statins (Table 4). For comparison purposes, we included these variables in the multiple regression models for transformed values of CTAs and total CAs, even though the influence of some of these variables was clearly not significant. For all three variables—CTA_t_, CSA_t,_ and CA_t_—the most significant confounding variable was the duration of disease (*p* = 0.0042, *p* = 0.00003, and *p* = 0.00002, respectively), showing a positive relationship between the number of aberrations and years that have passed since the diagnosis. No other variables had a statistically significant influence on CTA_t_. The whole model explained 11.1% of the total variability in CTA_t_ values. For CSA_t_, BMI was the next important confounding variable, showing a significant positive relation between increased body mass and transformed frequency of chromosome-type aberrations (*p* = 0.0105). In addition, the use of statins showed a negative relation with borderline significance (*p* = 0.0763). The whole model explained as much as 29.5% of the total variability in CSA_t_ values. Finally, the whole model for CA_t_ showed similar tendencies as a model for CSA_t_, but with lower statistical significance and lower percentage (25.8%) of total variability explained.

Interestingly, the stepwise multiple regression procedure excluded the age of the subjects from the model despite the positive correlation found in univariate analysis. As mentioned above, age was also correlated with other variables, particularly with duration of disease. We performed partial regression analysis using the transformed frequency of CSAs and found that the partial correlation between CSA_t_ and duration of disease was still significant after accounting for the influence of age (r = 0.265, *p* = 0.022), while the partial correlation between CSA_t_ and age became non-significant (r = 0.189, *p* = 0.1032) after accounting for the influence of duration of disease.

## 4. Discussion

The main goal of our study was to compare different types of chromosome aberrations in groups of patients with T1DM, T2DM, and control subjects without diabetes.

The total frequency of chromosome aberrations in lymphocytes of control subjects in this study (1.83/100 cells) is in good agreement with our historical control data (form 1.68 to 2.11 per 100 cells) [26,27].

Both T1DM and T2DM had a higher number of all types of aberrations than controls. These results align with other studies of T2DM subjects [28,29,30], including smoking and non-smoking T1DM and T2DM subjects [31]. In contrast, other authors did not reveal any difference in CA number between controls and T1DM [32] and untreated, mostly recently diagnosed T2DM [16]. This might suggest that the type and duration of diabetes could impact the CA number. Nevertheless, our study quite clearly indicated that the frequency of both CTAs and CSAs depends on the duration of the disease, not the type of diabetes. Interestingly, the same type of association has been reported in one CA [28] and several MN studies in T2DM patients [33,34,35]. Conversely, other studies failed to find a correlation between the frequency of MN and disease duration in T2DM [36,37]. No correlation between the frequency of MN and duration of T1DM was found [35,38]. It is worth noting that the average diabetes duration in these studies was significantly shorter than ours, with an average disease duration of 13 years for both types. Thus, we might hypothesize that shorter disease duration was insufficient to show a significant correlation.

In our study, T2DM patients had a higher number of CAs and CSAs than T1DM patients. To the best of our knowledge, there is only one study comparing CAs between T1DM and T2DM [31] where the authors showed a higher number of CAs in T2DM than in T1DM, but the significance of the difference was not presented. A higher number of CAs and CSAs in T2DM might be related to a higher prevalence of other comorbidities, such as obesity, arterial hypertension, and dyslipidemia, characteristic of T2DM. There are accumulating data that these comorbidities, especially obesity, are related to chronic inflammation, oxidative stress, consequent DNA damage, and increased cancer risk [39,40,41]. Indeed, we found that BMI is a significant risk factor for chromosome-type aberrations in a multivariate model. The same trend was observed in some studies of obese subjects using an MN assay [42,43,44]. Additionally, impaired DNA repair capacity [45,46] and increased DNA damage [39] in obese subjects were reported in several studies. The correlation between CSAs and BMI of patients with diabetes in our study aligns with these data. Interestingly, one study found a paradoxically lower frequency of MN alongside with a lower relative risk of developing lung cancer in men with a BMI ≥25 kg/m^2^ compared with normal-weight men [47]. However, most of the subjects in this study group are occupationally exposed to polycyclic aromatic hydrocarbons (PAHs), and the authors are trying to explain the reverse association between BMI and both chromosome damage and lung cancer risk by more active metabolism of PAHs in over-weight individuals.

Some authors revealed increased CAs or MN levels in patients with diabetic neuropathy [30,34] or nephropathy [33]. We did not find any association between the frequency of CAs and diabetes complications, but our sample size was small and, consequently, the statistical power of such analysis was quite low. Similarly, we did not find any influence of antidiabetic drugs (insulin and metformin) on the frequency of CAs in patients with diabetes. However, in the multivariate model, we found that lower levels of chromosome damage, mostly of chromosome-type aberrations, might be probably associated with the use of statins. Although this association is statistically non-significant (*p* = 0.0763), there are some literature data that may support a protective effect of statins against DNA damage and prompt further investigations in this field. For example, long-term therapy with simvastatin lowered levels of DNA damage in the lymphocytes of dyslipidemic T2DM patients, as measured using the Comet assay [48]. A negative correlation between the use of statins and results of both Comet and MN assays was found in univariate analysis; however, this association was no longer seen in the lymphocytes of patients with heart failure after multivariate analysis [49]. Additionally, simvastatin treatment in vitro lowered the frequency of sister-chromatid exchange in the lymphocytes of hemodialyzed patients [50]. In addition, atorvastatin, rosuvastatin [51], and simvastatin [52] were able to reduce the number of somatic mutations in *Drosophila melanogaster*, induced by doxorubicin, an antineoplastic drug capable of generating reactive oxygen species.

One previous study using FISH technology reported unusually high numbers of stable chromosome aberrations (translocations and inversions) in patients with diabetes [29] and linked their appearance to a higher risk of death. Using conventional Giemsa staining of chromosomes, we were unable to confirm this finding. However, we found a stronger association of chromosome-type aberrations with the duration of diabetes and BMI compared with chromatid-type aberrations. It is well-established that, in peripheral lymphocytes, CSAs are formed in vivo, whereas CTAs occur only in vitro from preexisting DNA lesions [13]. Thus, higher frequencies of CSAs, especially in individuals with more prolonged disease duration, may reflect actual accumulation of chromosomal damage not only in their lymphocytes, but also in other cells. Finding higher frequencies of CSAs in patients with diabetes may link epidemiological studies showing higher cancer risk in these patients [6,8] with our previous studies showing higher cancer risk in subjects with higher rates of chromosome-type aberrations [13,14]. Although this is not a causal link, it is highly probable that both associations might have the same mechanism. Chronic inflammation seems to be an excellent candidate for this mechanism. On the other hand, increased frequencies of CTAs may reflect the accumulation of primary DNA damage in lymphocytes arising due to oxidative stress and impaired excision repair [39,46]. It cannot be excluded that this type of damage may be related to the development of different diabetes complications and comorbidities.

In conclusion, we found that both T1DM and T2DM patients had a higher number of all types of aberrations than controls, which increases with the prolonged disease duration. A higher body mass index was associated with a higher frequency of chromosome-type aberrations. The use of statins might be beneficial for reducing chromosome damage, but further investigations are needed to establish this association.

## Figures and Tables

**Figure 1 life-13-01926-f001:**
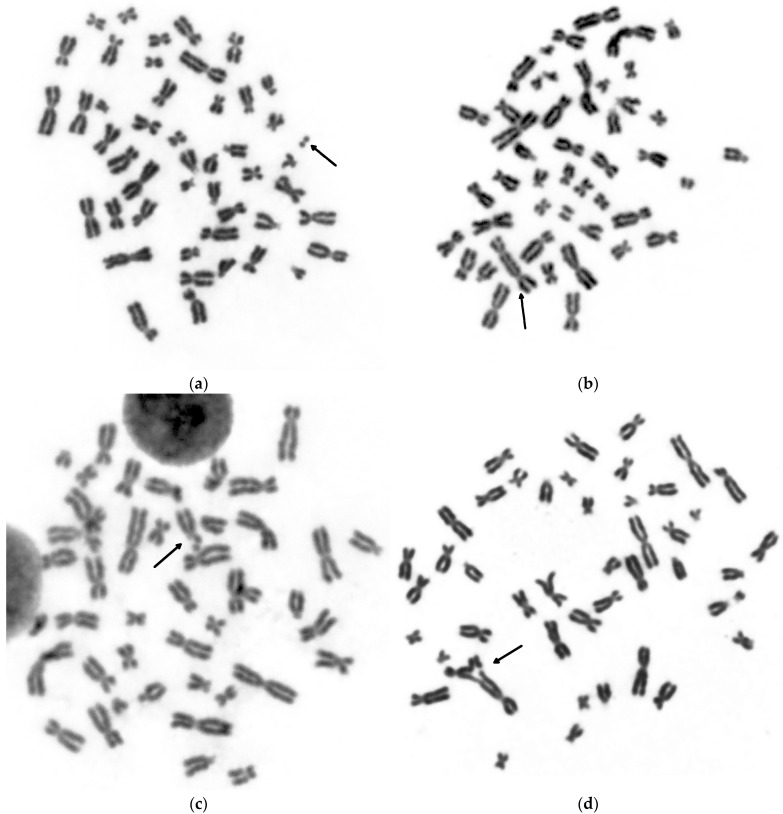
Representative images showing metaphases with different types of chromosome aberrations: (**a**) chromosome-type aberration—acentric fragment; (**b**) chromosome-type aberration—dicentric chromosome with accompanying fragment; (**c**) chromatid-type aberration—chromatid break; (**d**) chromatid-type aberration—triradial. All chromosome aberrations are indicated by arrows.

**Figure 2 life-13-01926-f002:**
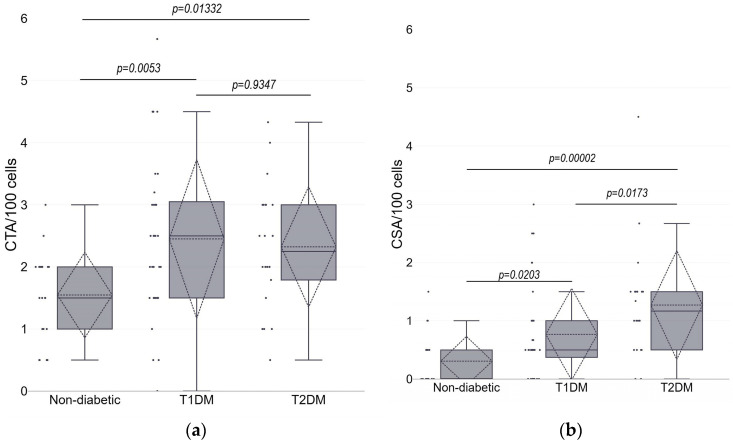
The frequency of chromatid-type (CTA, panel (**a**)) and chromosome-type (CSA, panel (**b**)) aberrations in lymphocytes of subjects without diabetes and patients with type 1 (T1DM) and type 2 (T2DM) diabetes. Box plots show the first quartile, median, and third quartile; the end whiskers mark the 10th and 90th percentiles. The dashed line shows average value, dashed triangles—standard deviation values. Black dots represent individual values. *p* values were calculated using Man-Whitney U-test.

**Figure 3 life-13-01926-f003:**
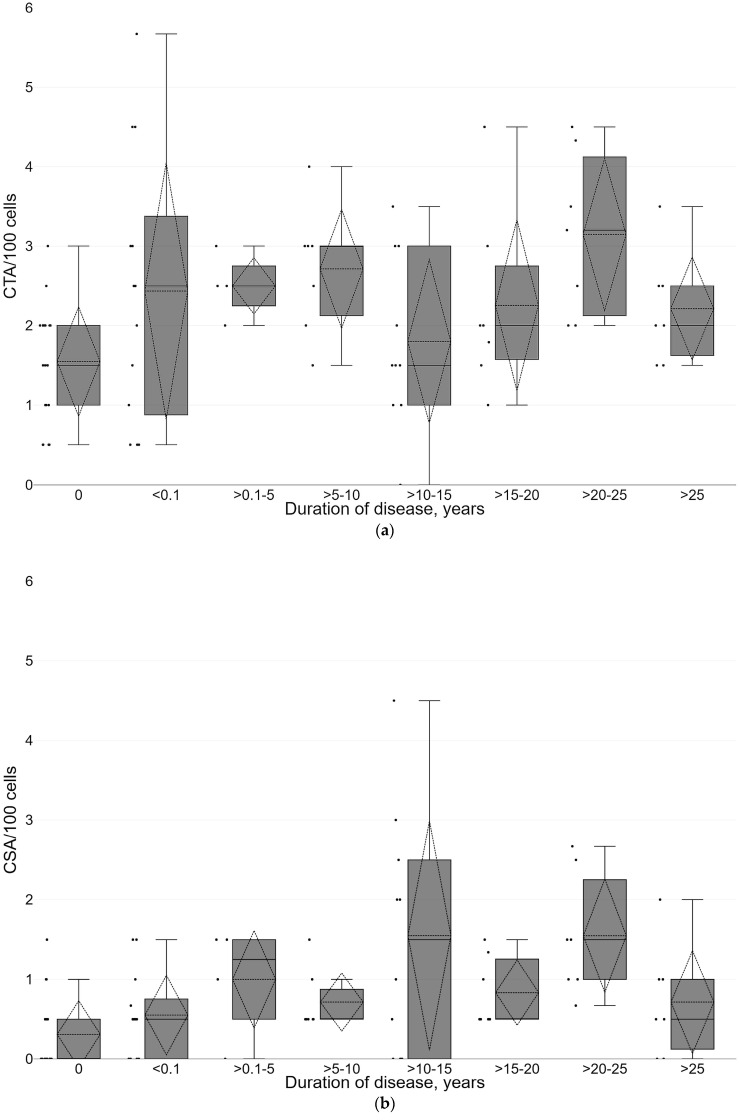
The frequency of chromatid-type (CTA, panel (**a**)) and chromosome-type (CSA, panel (**b**)) aberrations in lymphocytes of patients with diabetes grouped according to the duration of disease (time after diagnosis). Box plots show the first quartile, median, and third quartile; the end whiskers mark the 10th and 90th percentiles. The dashed line shows the average value and dashed triangles—standard deviation values. Black dots represent individual values.

**Figure 4 life-13-01926-f004:**
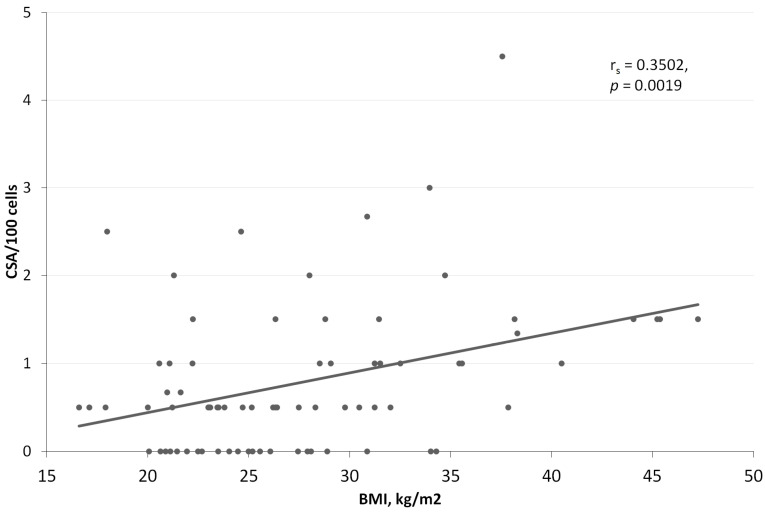
Correlation between body mass index (BMI) and number of chromosome-type aberrations (CSA) in lymphocytes of patients with diabetes and controls. The dashed line shows a linear regression fit.

**Table 1 life-13-01926-t001:** Demographic and clinical data for the different groups of patients with diabetes and controls.

Variable		Diabetes				Controls		*p* Values		
		Type 1		Type 2				Type 1 vs. Type 2	Type 1 vs. Controls	Type 2 vs. Controls
		N	%	N	%	N	%			
Demographic data										
Sex	Female Male	13 20	39.4 60.6	13 9	59.1 40.9	17 4	81.0 19.0	0.152	0.003	0.119
Smoking	Nonsmoker Former/Current smoker	15 18	45.5 54.5	16 6	72.7 27.3	15 6	71.4 28.6	0.046	0.061	0.924
Age in years, mean (SD)		39.8 (15.4)		61.8 (14.1)		38.1 (13.6)		0.000	0.809	0.000
BMI, kg/m^2^, mean (SD)		23.6 (3.7)		35.5 (6.1)		25.9 (4.6)		0.000	0.181	0.000
				Diabetes and complications						
Time after diagnosis in years, mean (SD)		13.5 (13.3)		12.9 (7.7)				0.672		
Family history of diabetes	No Yes	19 14	57.6 42.4	5 17	22.7 77.3	15 6	71.4 28.6	0.011	0.304	0.001
Nephropathy	No Yes	26 7	78.8 21.2	18 4	81.8 18.2			0.783		
Polyneuropathy	No Yes	22 11	66.7 33.3	10 12	45.5 54.5			0.118		
Autonomic heart neuropathy	No Probable Yes ND	21 6 4 2	63.6 18.2 12.1 6.1	9 11 1 1	40.9 50.0 4.5 4.5			0.091		
Concomitant diseases										
Ischemic heart disease	No Yes	31 2	93.9 6.1	14 8	63.6 36.4	21 0	100.0 0.0	0.004	0.250	0.002
Arterial hypertension	No Yes	20 13	60.6 39.4	2 20	9.1 90.9	19 2	90.5 9.5	0.000	0.017	0.000
Dyslipidemia	No Yes	16 17	48.5 51.5	3 19	13.6 86.4	19 2	90.5 9.5	0.008	0.002	0.000
Thyroid diseases (excl. cancer)	No Yes	24 9	72.7 27.3	13 9	59.1 40.9	16 5	76.2 23.8	0.291	0.777	0.232
Use of prescribed medications										
Insulin	No Yes	7 26	21.2 78.8	7 15	31.8 68.2			0.376		
Metformin	No Yes			4 18	18.2 81.8					
Statins	No Yes	23 10	69.7 30.3	12 10	54.5 45.5	19 2	90.5 9.5	0.252	0.073	0.009
Antihypertensive drugs	No Yes	20 13	60.6 39.4	2 20	9.1 90.9	19 2	90.5 9.5	0.000	0.017	0.000

**Table 2 life-13-01926-t002:** Frequency of chromosome aberrations per 100 cells in different groups of patients with diabetes and controls.

Groups	Aberrations per 100 Cells *														
	CTB			CTE			CSB			CSE			Total CA		
	Mean	SD	Min–Max	Mean	SD	Min–Max	Mean	SD	Min–Max	Mean	SD	Min–Max	Mean	SD	Min–Max
Non-diabetic	1.48	0.68	0.50–3.00	0.07	0.18	0.00–0.50	0.21	0.29	0.00–1.00	0.10	0.22	0.00–0.67	1.83	0.90	0.50–4.00
T1DM	2.38	1.26	0.00–5.33	0.08	0.18	0.00–0.50	0.39	0.53	0.00–2.00	0.37	0.51	0.00–2.00	3.13	1.47	1.00–6.00
T2DM	2.26	0.93	0.50–4.00	0.06	0.16	0.00–0.50	0.80	0.53	0.00–2.00	0.47	0.65	0.00–2.50	3.60	1.47	1.00–7.50
*p*-value ****	0.0059			0.8981			0.0002			0.0503			0.0002		

* CTB—chromatid breaks; CTE—chromatid exchanges; CSB—chromosome breaks; CSE—chromosome exchanges; total CA—total number of aberrations.** Kruskal–Wallis test.

**Table 3 life-13-01926-t003:** Distribution of chromosome aberrations in the studied groups of individuals according to their categorical demographic and clinical variables.

Variables *		Diabetes						Controls		
		Type 1			Type 2					
		CTA (SD)	CSA (SD)	CA (SD)	CTA (SD)	CSA (SD)	CA (SD)	CTA (SD)	CSA (SD)	CA (SD)
Sex	Female Male	2.32 (1.28) 2.53 (1.34)	0.97 (0.98) 0.63 (0.65)	3.21 (1.54) 3.08 (1.45)	2.41 (1.05) 2.20 (0.95)	1.13 (0.77) 1.48 (1.20)	3.54 (1.65) 3.68 (1.27)	1.56 (0.75) 1.50 (0.58)	0.24 (0.36) 0.63 (0.63)	1.76 (0.87) 2.13 (1.11)
Smoking	NS CS/FS	2.45 (1.12) 2.45 (1.47)	0.84 (0.88) 0.70 (0.75)	3.28 (1.68) 3.01 (1.30)	2.35 (0.96) 2.25 (1.17)	1.13 (0.69) 1.67 (1.47)	3.48 (1.47) 3.92 (1.56)	1.73 (0.62) 1.08 (0.74)	0.37 (0.48) 0.17 (0.26)	2.07 (0.84) 1.25 (0.82)
Thyroid diseases	No Yes	2.49 (1.42) 2.36 (0.99)	0.76 (0.83) 0.80 (0.75)	3.13 (1.55) 3.13 (1.29)	2.39 (1.17) 2.22 (0.71)	1.50 (1.08) 0.94 (0.68)	3.89 (1.58) 3.17 (1.27)	1.50 (0.55) 1.70 (1.15)	0.25 (0.41) 0.50 (0.50)	1.72 (0.82) 2.20 (1.15)
Family history of DM	No Yes	2.37 (1.23) 2.56 (1.43)	0.95 (0.98) 0.52 (0.35)	3.21 (1.54) 3.02 (1.41)	2.32 (1.43) 2.32 (0.88)	1.40 (0.81) 1.24 (1.02)	3.73 (1.98) 3.56 (1.37)	1.70 (0.68) 1.17 (0.68)	0.27 (0.46) 0.42 (0.38)	1.97 (0.90) 1.50 (0.89)
AH	No Yes	2.61 (1.59) 2.21 (0.65)	0.78 (0.86) 0.74 (0.72)	3.26 (1.68) 2.94 (1.09)	2.50 (0.71) 2.31 (1.03)	1.00 (0.71) 1.30 (0.99)	3.50 (1.41) 3.61 (1.52)	1.55 (0.66) 1.50 (1.41)	0.32 (0.45) 0.25 (0.35)	1.84 (0.91) 1.75 (1.06)
Dyslipidemia	No Yes	2.79 (1.60) 2.13 (0.87)	0.82 (0.96) 0.72 (0.64)	3.45 (1.73) 2.83 (1.14)	1.93 (0.12) 2.39 (1.01)	1.11 (0.54) 1.30 (1.02)	3.04 (0.51) 3.68 (1.57)	1.55 (0.66) 1.50 (1.41)	0.32 (0.45) 0.25 (0.35)	1.84 (0.91) 1.75 (1.06)
IHD	No Yes	2.48 (1.33) 2.00 (0.71)	0.74 (0.79) 1.25 (1.06)	3.14 (1.46) 3.00 (2.12)	2.27 (0.92) 2.42 (1.17)	0.99 (0.61) 1.77 (1.27)	3.26 (1.29) 4.19 (1.67)	N.A.	N.A.	N.A.
Nephropathy	No Yes	2.51 (1.42) 2.21 (0.76)	0.83 (0.87) 0.52 (0.41)	3.24 (1.56) 2.74 (1.03)	2.41 (1.03) 1.95 (0.82)	1.12 (0.70) 1.96 (1.70)	3.53 (1.56) 3.91 (1.15)	N.A.	N.A.	N.A.
PNP	No Yes	2.64 (1.52) 2.06 (0.56)	0.80 (0.84) 0.70 (0.75)	3.33 (1.62) 2.74 (1.06)	2.56 (1.11) 2.13 (0.88)	1.30 (0.62) 1.25 (1.20)	3.86 (1.50) 3.38 (1.48)	N.A.	N.A.	N.A.
CAN	No Yes MD	2.54 (1.47) 2.35 (1.06) 2.00 (0.71)	0.87 (0.91) 0.47 (0.38) 1.25 (1.06)	3.30 (1.57) 2.82 (1.20) 3.00 (2.12)	2.06 (0.98) 2.64 (0.92) 1.00	1.39 (1.27) 1.25 (0.72) 0.50	3.44 (1.33) 3.89 (1.53) 1.50	N.A.	N.A.	N.A.
Insulin	No Yes	2.52 (1.89) 2.43 (1.14)	0.38 (0.39) 0.87 (0.85)	2.74 (1.70) 3.24 (1.41)	2.50 (1.08) 2.24 (0.98)	1.29 (0.57) 1.27 (1.11)	3.79 (1.32) 3.51 (1.58)	N.A.	N.A.	N.A.
Metformin	No Yes	N.A.	N.A.	N.A.	2.13 (1.18) 2.37 (0.98)	1.38 (0.25) 1.25 (1.06)	3.50 (1.08) 3.62 (1.57)	N.A.	N.A.	N.A.
Statins	No Yes	2.66 (1.44) 1.97 (0.76)	0.88 (0.85) 0.52 (0.63)	3.40 (1.53) 2.52 (1.16)	2.47 (1.15) 2.15 (0.78)	1.21 (0.65) 1.35 (1.27)	3.68 (1.60) 3.50 (1.39)	1.55 (0.66) 1.50 (1.41)	0.32 (0.45) 0.25 (0.35)	1.84 (0.91) 1.75 (1.06)
Antihypertensive drugs	No Yes	2.56 (1.59) 2.28 (0.69)	0.83 (0.88) 0.67 (0.69)	3.26 (1.68) 2.94 (1.09)	2.50 (0.71) 2.31 (1.03)	1.00 (0.71) 1.30 (0.99)	3.50 (1.41) 3.61 (1.52)	1.55 (0.66) 1.50 (1.41)	0.32 (0.45) 0.25 (0.35)	1.84 (0.91) 1.75 (1.06)

* CAN—cardiac autonomic neuropathy; IHD—ischemic heart disease; PNP—peripheral neuropathy; AH—arterial hypertension; NS—nonsmoker; CS/FS –current/former smoker; N.A.—not applicable.

**Table 4 life-13-01926-t004:** Results of multiple regression of transformed frequency of chromosome aberrations per 100 cells on the most significant variables—duration of disease, body mass index (BMI), and use of statins.

Variables	Chromatid-Type Aberrations (CTA)			Chromosome-Type Aberrations (CSA)			Total Aberrations (CA)		
	Regression Coefficient	Standard Error	*p*	Regression Coefficient	Standard Error	*p*	Regression Coefficient	Standard Error	*p*
Intercept	1.7372	0.5614	0.0028	−1.8857	0.5963	0.0023	0.7105	0.5762	0.2216
BMI	−0.0882	0.1723	0.6104	0.4810	0.1831	0.0105	0.2608	0.1769	0.1448
Duration of disease	0.1654	0.0560	0.0042	0.26348	0.0595	0.00003	0.2609	0.0575	0.00002
Use of statins	−0.1947	0.1372	0.1604	−0.26218	0.145782	0.0763	−0.2421	0.1409	0.0899
Model properties									
Model *	CTA_t_ = 1.737191 − 0.0882006 Ln(BMI) + 0.165427 Ln(Duration of disease) − 0.19466 Ln(Use of statins)			CSA_t_ = 0.15173 × (BMI)^0.480953^ × (Duration of disease)^0.26338^ × (Use of statins)^−0.26218^			CA_t_ = 0.710502 + 0.260791 Ln(BMI) + 0.260852 Ln(Duration of disease) − 0.24212 Ln(Use of statins)		
R-square	0.1106			0.2950			0.2576		
Coefficient of multiple correlation	0.3326			0.5432			0.5076		
Goodness of fit	F_(3,72)_ = 2.9855, *p* = 0.0367			F_(3,72)_ = 10.0445, *p* = 0.00001			F_(3,72)_ = 8.3290, *p* = 0.00008		
Residual normality (Shapiro–Wilk test)	*p* = 0.4076			*p* = 0.1005			*p* = 0.6373		

* BMI is a continuous variable expressed as kg/m^2^; duration of the disease is an indicator variable, where 1 = no diabetes, 2 = new diagnosis, 3 = duration <5 years, 4 = duration >5–10 years, 5 = duration >10–15 years, 6 = durations >15-20 years, 7 = duration >20–25 years, 8 = duration >25 years; use of statins is an indicator variable where 1 = non-users and 2 = users; CTA_t_, CSA_t_, and CA_t_ are transformed (*Y* = 0.5[(*X*)^0.5^ + (*X* + 1)^0.5^]) frequency per 100 cells of chromatid-type, chromosome-type, and total number of aberrations, respectively.

## Data Availability

The data presented in this study are available on reasonable request from the corresponding author. The data are not publicly available due to restrictions set by the Ethics Committee.

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
