# Peer review of "Influence of Body Mass Index and Duration of Disease on Chromosome Damage in Lymphocytes of Patients with Diabetes"

_life, 2023, doi:10.3390/life13091926_

Round 1
Reviewer 1 Report
Comments to the manuscript:
Influence body mass index and of duration of disease on chromosome damage in lymphocytes of patients with diabetes
In this study, authors analyze chromosome aberrations (CA) in PBL from patients with Diabetes Mellitus, T1DM and T2DM and control subjects. They recorded the different types of chromosome aberrations such as chromatid breaks, chromatid exchanges, chromosome breaks, and chromosome exchanges and search for their association with different demographic and clinical parameters.
This reviewer considers that although the subject of DNA damage in patients with Diabetes has been explored for a long time, the present manuscript provides information on a gap in chromosomal studies in this disease.
However, I believe that the manuscript has some observations that it is important to address before being accepted for publication and that I mention below:
1. In the methodology section: In relation to the regression models, it is understandable why they unified the values since they are different units in each variable studied, but needs to explain/justify why they only show the model of the combination BMI+disease duration+use of statins; the use of statins does not show to be statistically significant in relation to the regression coefficient. In fact, without this explanation, it may be thought that other associations could also be important, such as the use of antihypertensive drugs, for example. This item can be explained in the methodology and put the data of the other models (if other models were made) as supplementary.
2. Table 1. Table 1 has values that do not agree with the text. It is evident that in the line corresponding to Metformin, there are errors. "Antihy-pertensive" and other words in that column should also be corrected.
3. P4, Line 156. Is CSE instead CSA.
4. Table 2. It is not clear why authors explain “ CTA – chromatid-type aberrations; CSA – chromosome-type aberrations”, and in the table they show specific types of aberrations that have no explanation.
5. In Results and discussion, Authors mention that that “the frequency of both CTA 261 and CSA depends on the duration of the disease” and CTAs
6. P9. The second paragraph of the Discussion, seems repetitive: 4 “however”, and “To the best of our knowledge, there are no studies comparing CA between T1DM 271 and T2DM. Anand et al. [31] showed a higher number of CA in T2DM than in T1DM, but 272 the significance of the difference was not presented…”
7. P9, L271-273. “To the best of our knowledge, there are no studies comparing CA between T1DM 271 and T2DM. Anand et al. [31] showed a higher number of CA in T2DM than in T1DM, but 272 the significance of the difference was not presented”. Is contradictory
8. CTAs are practically forgotten in their discussion, being that they are the majority. CSAs are related to cancer risk and it is very valid all studies performed to detect if there is a relationship with increased risk of cancer. But at no point in the manuscript do the authors explain why they leave behind the CTAs, these are important to explain DNA damage that may not have been processed, but they are there. So, perhaps they are leaving out an important fact, remember that in Diabetes it is not only important the risk of cancer, there are other clinical conditions that could have to do with CTAs ( ?) and these results are simply left aside.
9. On the other hand, an important part of their discussion is dedicated to statins, however, a significant p is not shown with any of the types of aberrations. Without clear explanations, it appears that authors are forcing their results toward making statin use significant in relation to DNA damage.
Author Response
We are very thankful for your valuable comments and suggestions. Please see attachment with our response.

Reviewer 2 Report
In this paper, the Authors present clearly an apparent increase in chromosome aberrations (CAs) in T1DM and T2DM patients compared to the control group. Interestingly the increased frequency of CAs depends on the duration of the disease and not the type of diabetes. The study is well-designed and results are presented in a clear way. Authors should include many representative images of metaphases depicting different kinds of CAs.
A careful proofreading will help improving the manuscript!
Author Response
We are very thankful to the reviewer for valuable comments and suggestion. Below is our response to the reviewer’s criticism.
In this paper, the Authors present clearly an apparent increase in chromosome aberrations (CAs) in T1DM and T2DM patients compared to the control group. Interestingly the increased frequency of CAs depends on the duration of the disease and not the type of diabetes. The study is well-designed and results are presented in a clear way. Authors should include many representative images of metaphases depicting different kinds of CAs.
Response. Thank you for your suggestion. A figure (Fig. 1 in corrected version) containing images of different kinds of CAs is included now into the manuscript.
Round 2
Reviewer 1 Report
In V2, The authors have modified and corrected the manuscript according to the recommendations.